# What we think prayers do: Americans' expectations and valuation of intercessory prayer

**Linda Thunström**[1]*, **Shiri Noy**[2]

1 Department of Economics, University of Wyoming, Laramie, Wyoming, United States of America,
2 Department of Anthropology and Sociology, Denison University, Granville, Ohio, United States of America

* lthunstr@uwyo.edu

**Data Availability Statement:** All data and code files are publicly available from openicpsr https://www.openicpsr.org/openicpsr/project/163981/version/V1/view.

## Abstract

Praying for others in the wake of a disasters is a common interpersonal and public response to tragedy in the United States. But these gestures are controversial. In a survey experiment, we elicit how people value receiving a prayer from a Christian stranger in support of a recent hardship and examine factors that affect the value of the prayer. We find that people who positively value receiving the prayer do so primarily because they believe it provides emotional support and will be answered by God. Many also value the prayer because they believe it will improve their health and wealth, although empirical support of such effects is lacking. People who negatively value receiving the prayer do so primarily because they believe praying is a waste of time. The negative value is particularly large if people are offended by religion. Finally, the hardship experienced by the prayer recipient matters to the intensity by which recipients like or dislike the gesture, suggesting the benefit of prayers varies not only across people, but also across contexts.

## Introduction

Asking for prayers during hardship and sending prayers to others are common responses to both personal difficulties and public disasters in the United States. For instance, President Trump proclaimed March 15, 2020, a National prayer day in response to the COVID-19 pandemic, while President Obama publicly expressed that he and first lady Michelle Obama were praying for the victims of the mass shooting in Las Vegas in 2017 [1, 2].

However, this response is deeply polarizing. Many Americans actively seek out others' prayers in times of hardship; In addition to asking for prayers from friends and family, people may request, or even pay for, prayers [3, 4]. Others, however, believe that prayers are either useless or potentially even harmful in that they may detract from material help. In particular, the value of receiving prayers is highly divided across religious belonging and beliefs [5]. Previous research finds that religious Christians are willing to give up money for a prayer, while atheists and agnostics are willing to pay *not* to receive a prayer. It is, however, unknown what explains prayer recipients' positive or negative values of these gestures, i.e., *why* do religious Christians

**Funding:** The author(s) received no specific funding for this work.

**Competing interests:** The authors have declared that no competing interests exist.

value receiving a prayer, and why are non-believers (i.e., atheists and agnostics) averse to receiving the same prayer?

Better understanding what determines people's preferences for receiving prayers is important for several reasons. First, receiving prayers from others in times of hardship may affect behavior, especially if prayers are believed to generate real benefits, such as health and wealth. For instance, people react less to information about catastrophes when they believe they have God's support [6], which might affect coping mechanisms to mitigate risk. Second, a better understanding of people's motives for valuing or disliking prayers in times of hardship may increase the respect and tolerance for people's preferences for this gesture [7]. This, in turn, helps improve the targeting of prayers, enabling better help to those in need. Third, knowledge about people's perceptions of the benefit of prayers may help explain the mechanisms behind why praying for others may crowd out material help [8].

While the determinants of the value of prayers from others are largely unknown, results from studies on benefits and expectations from prayers *by self* may provide guidance. Religiosity increases in response to disasters [9, 10], suggesting that religious engagement, through comradery or rituals such as prayers, are perceived as helpful in hard times. Further, the act of praying may help mitigate hardships. First, praying generates emotional comfort [11, 12]. Second, a common motive for praying is to ask God for help with personal health or finances [13], suggesting that there is some expectation of prayers being answered by God. The belief that God may directly intervene and affect wealth and health is particularly salient in Protestant prosperity theology [14].

If prayers are perceived to have benefits beyond the personal experience of praying, such as comfort from a shared experience, it seems possible that the knowledge of being prayed for *by others* could similarly increase emotional comfort. Also, if people expect the act of praying to bring emotional comfort to the person who prays, they may value the emotional benefit to the *sender*, suggesting an altruistic motive for valuing prayers from others. Further, it seems plausible that prayers from others could be expected to help health and wealth issues, similar to the expected benefits from prayers conducted by oneself.

Guidance from previous research on potential causes of negatively valuing prayers from others is more limited. Prayer aversion is prevalent amongst non-believers [5]. One possibility is that prayers are regarded as offensive because they represent religion as an institution. Further, discrimination of atheists in the U.S. is well-documented [15–17], which may cause resentment amongst non-believers towards the Christian majority. Non-believers may also perceive praying to be a useless activity (a "waste of time"), using up resources (e.g., time, effort) that could have been used more productively.

To examine what causes people to value or dislike receiving prayers from others in the wake of a hardship, we designed an experimental survey to elicit the monetary value (willingness-to-pay; WTP) to religious Christians and non-believers of receiving a prayer from a Christian stranger. We asked about their reasons for stating positive or negative values. The experiment was incentivized, with the intent to elicit truthful values of the intercessory prayer. Given the novelty of the topic, an important part of the study design was an open ended component, where participants were given the opportunity to briefly offer reasons for their values of prayers beyond those pre-specified by the authors.

## Materials and methods

We recruited 656 survey participants across the U.S. via the survey company Qualtrics. The survey lasted around 15 minutes and participants received the regular payment from Qualtrics as well as an additional USD5 in e-currency that they could spend to solicit or avoid prayers

from a Christian stranger in the experiment. The recruitment cost per participant is higher when recruiting via Qualtrics compared to other online survey platforms, such as Amazon Mechanical Turk, but the advantage of using Qualtrics is their data quality checks, which reduce the prevalence of data quality issues [18]. For instance, to ensure validity of the responses and avoid duplication, Qualtrics checks IP address of all responses and uses digital fingerprinting technology. Qualtrics replaces respondents that fail any attention checks, as defined by the researcher, or in other ways appear fraudulent (this is evaluated in collaboration with the researcher), as well as respondents who seem to rush through the survey, i.e., completes the survey in less than half the median survey completion length. Verification of responder identities is generally done by Qualtrics' sample partners and include TrueSample, Verity, SmartSample, panelist ID number, cookies, Geo-IP address, LinkedIn information comparison, and digital fingerprinting.

We limited our sample to Christians and atheists/agnostics to facilitate cross-group comparisons with previous literature. Christians are the majority religious group in the U.S. (65 percent of the population identify as Christians), while the group of non-believers is fast growing and make up around 10 percent of the population [19]. Given some of our participants opted to receive a prayer from a stranger as part of our study, as described below, we also recruited senders of prayers. However, their role was solely to read the short, anonymous, statements of hardships of our participants and send them supporting prayers. We did not elicit any information from the senders and when discussing our sample, we refer only to the recipients of prayers.

Based on previous results [5], we expected religious Christian participants to positively value prayers and non-believers to negatively value prayers. To ensure enough power to statistically detect meaningful differences within the religious Christian group, we recruited more Christians than non-believers. Specifically, we asked Qualtrics to recruit 482 Christians (screening: identify as Protestants or Catholics and state that they believed in God) and 174 non-believers (screening: identify as atheists or agnostics and stated that they deny or are uncertain of God's existence). Of Christians in our study, 196/482 were Catholics and 286/482 were Protestants. Of non-believers, 15/174 were atheists and 159/174 were agnostics. The study was approved by the University of [blinded] (#20200306LT02694) and [blinded] University (SP20 #29) Institutional Review Boards.

The sequence of the experimental survey was as follows: Step 1: Participants stated their consent to participate and to commit to thoughtfully provide their best answers. Step 2: Participants answered background questions such as gender, age, religious identity, belief in God, state of residency. Since some of these questions were used to screen Christians and non-believers they needed to be at the front end of the survey. Step 3: Participants completed a brief training on the mechanism (multiple-price-list; MPL) used to elicit the value (WTP) of prayers in Step 6 below. Step 4: Participants were asked to describe (using max 500 words) a recent hardship. Because the experimental survey was fielded in May 2020, i.e., during the COVID-19 pandemic, we also asked questions to account for differences in pandemic experiences. Participants were therefore asked whether they, or a loved one, had been negatively affected by COVID-19, and if they answered "yes" to that question, they were asked to describe the COVID-19 related hardship, otherwise they were asked to describe any other hardship they had experienced in the last couple of years. They were asked not to include any personal identifiers in their description. They were asked to state how difficult it was to deal with the described hardship emotionally or practically, and how they would characterize the hardship (health, relationship, financial, for themselves or for a loved one). Step 5: Participants were informed that they would be offered the opportunity to receive a supportive prayer from a Christian stranger who believes in God, aimed at the positive and peaceful resolution of the

hardship they had described in Step 4. To prevent financial altruism towards the Christians stranger from affecting the participants' value for prayers, participants were informed that the stranger's compensation would be completely independent of their choices in the survey. Step 6: Following previous studies [5, 20], we used a MPL to elicit the WTP for receiving an intercessory prayer in support of the hardship described in Step 4. In short, participants were informed they were endowed with e-currency corresponding to USD$5 in financial support of the positive and peaceful resolution of their hardship. Some or all of this money could be used in exchange for securing a supportive prayer from the stranger, or preventing such a prayer from being undertaken on their behalf. Depending on their answers to the questions in this part of the survey, participants' values (WTP) of the prayer could range from negative to positive. Step 7: Participants who stated a positive/negative value for an intercessory prayer from the Christian stranger were asked, in an open ended question, to briefly specify why they valued the prayer positively/negatively. They were also asked about their agreement/disagreement with a set of statements about factors that might have determined their value of the prayer. Further, participants who stated a positive value for the prayer were asked to state their belief about the probability (0–100) that the prayer from the stranger would be answered by God. Step 8: Participants were asked about their general attitudes towards prayers. They were asked about their level of religiosity, education, political preferences, income, conservatism (we measured both social and economic conservatism per the SEC scale (SECs, [21]), as well as a single scale ranging from liberal and conservative), marital status and number of adults and children in their household, and thereafter the experimental survey ended. The survey instrument is deposited, as part of the Supplemental online material, in the open repository ICSPR at https://www.openicpsr.org/openicpsr/project/163981/version/V1/view. There, we also post the data and code used for all analysis in the manuscript.

Descriptive statistics of Christians and non-believers are shown in Table 1.

The MPL as a mechanism to elicit monetary values has benefits over alternative methods, such as experimental auctions. The MPL is relatively easy to understand and it is transparent to participants that stating truthful values in the study is in their best interest [22]. The MPL, however, also has drawbacks. First, it generates measures of participants' WTP stated as intervals rather than point values. Following previous studies [5, 20], we assign the interval midpoint as the value to participants. The end intervals of MPLs have no upper/lower limits. For end intervals, values must therefore be imputed. Our primary analysis uses the most conservative measure of WTP, imputing end values equal to -$5 and $5. Second, MPLs may generate internally inconsistent values. For instance, a participant is internally inconsistent if they indicate by their choices in the MPL that they are willing to forgo $3 to receive a prayer, but unwilling to forgo a smaller amount (e.g., $2). While the experimental survey can be coded to disable them, the inconsistencies may be important to note, given they may signal inattention, misunderstandings, etc., on the part of the participant. Our analysis of participants' WTP for prayers includes all participants whose WTP was internally consistent. While our sample has a total of 482 Christians and 174 non-believers, our analysis is based on the 451 Christian participants and the 166 nonbelievers who reported internally consistent WTP values.

## Results and discussion

### The value of a prayer

We find that Christians (N = 451) value prayers at an average of $2.34 while non-believers (N = 166) are willing to pay $1.56 *not* to be prayed for by a Christian stranger, see Fig 1. These values are consistent with previous results [5].

**Table 1. Descriptive statistics for Christians and non-believers.**

| Variable | Christians | | | Non-believers | | |
|---|---|---|---|---|---|---|
| | Obs | Mean | Std. Dev. | Obs | Mean | Std. Dev. |
| Female | 482 | 0.521 | 0.500 | 174 | 0.437 | 0.497 |
| Age | 482 | 64.861 | 10.179 | 174 | 56.483 | 16.345 |
| Affected by Covid-19 | 482 | 0.539 | 0.499 | 174 | 0.626 | 0.485 |
| Low SECs | 482 | 0.168 | 0.374 | 174 | 0.655 | 0.477 |
| Medium SECs | 482 | 0.313 | 0.464 | 174 | 0.270 | 0.445 |
| High SECs | 482 | 0.519 | 0.500 | 174 | 0.075 | 0.264 |
| College | 482 | 0.755 | 0.430 | 174 | 0.856 | 0.352 |
| Democrat | 482 | 0.305 | 0.461 | 174 | 0.557 | 0.498 |
| Republican | 482 | 0.517 | 0.500 | 174 | 0.236 | 0.426 |
| Other political party | 482 | 0.178 | 0.383 | 174 | 0.207 | 0.406 |
| Low income | 482 | 0.305 | 0.461 | 174 | 0.259 | 0.439 |
| Low/medium income | 482 | 0.394 | 0.489 | 174 | 0.368 | 0.484 |
| High/medium income | 482 | 0.197 | 0.398 | 174 | 0.190 | 0.393 |
| High income | 482 | 0.104 | 0.305 | 174 | 0.184 | 0.389 |
| Health issue | 482 | 0.301 | 0.459 | 174 | 0.299 | 0.459 |
| Financial issue | 482 | 0.295 | 0.456 | 174 | 0.345 | 0.477 |
| Relationship issue | 482 | 0.193 | 0.395 | 174 | 0.138 | 0.346 |
| Other type of issue | 482 | 0.212 | 0.409 | 174 | 0.218 | 0.414 |

Note: Low income: annual household income up to $50,000, low/medium income: $50,001-$100,000, high/medium income: $100,001-$150,000, high income: $150,001 and above. College is a binary variable that takes the value 1 if a participant has some college education. SECs is the conservatism scale developed by Everett (2013). The dummy variables Low SECs, Medium SECs and High SECs are created by splitting the total sample in 3 equal shares, labelling the third with the lowest SECs scores "Low SECs," and so on. Health issue is a binary variable that takes the value 1 if the participant described a hardship that constitutes a health issue (for self or a loved one); Financial issue is a binary variable that takes the value 1 if the hardship is a financial issue; Relationship issue is a binary variable that takes the value 1 if the hardship is a relationship issue; Other type of hardship is a binary variable that takes the value 1 if the participant experienced a hardship other than a health, financial or relationship issue for him-/herself or a loved one.

We do not find a statistically significant difference in WTP across Protestants and Catholics (a two sided $t$-test: t(449) = -0.283; p = 0.778; a Wilcoxon Mann Whitney test: z(449) = -.729; p = .466). A lower negative WTP for atheists ($-2.48, on average) than for agnostics ($-1.47, on average) is consistent with previous findings [5], but our sample of atheists (N = 15) is too small for the difference to be statistically significant, and this finding should be regarded as highly preliminary.

Of participants reporting internally consistent WTP values, 56 percent (348/617) had been negatively affected by COVID-19 and therefore communicated a COVID-19 hardship in our study. We did not find a difference between their WTP for the prayer and the value of those unaffected by COVID-19 (two sided t-test: t(617) = -.599; p = .549; Wilcoxon Mann Whitney test: z(617) = -.595; p = .552). We also examined what determines whether a person positively values the prayer from the Christian stranger. In short, people who are relatively young, conservative, have a low income, and Christian are more likely to assign a positive value to receiving the prayer (for details, see Supplemental Online Material).

## Determinants of the value of intercessor prayers

**Reasons people <u>positively</u> value prayers from religious strangers.** Participants who stated a positive WTP for receiving a prayer from a stranger (Christian: N = 375/451; non-believers: N = 56/166), were asked about the factors that contributed to the value of the prayer.

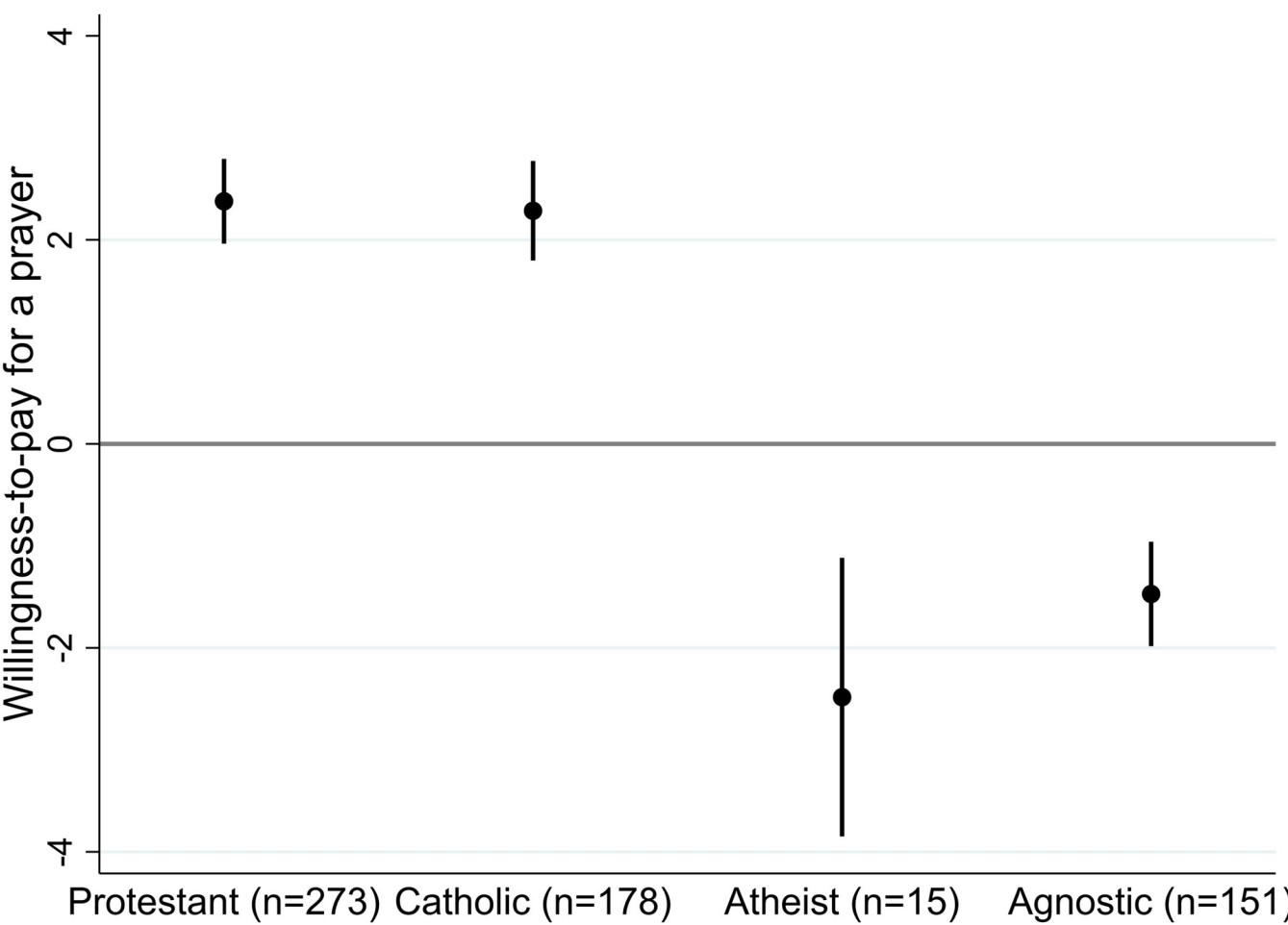

**Fig 1. Mean WTP (using most conservative WTP measure) for a prayer from a Christian stranger.** Note: Error bars show 95 percent confidence intervals.

Large majorities of both Christians and non-believers who value the prayer do so because it gives them emotional comfort to know that the stranger is thinking of them. The answers to the open ended question provide additional information on the comfort people experience from receiving the prayer–e.g., one non-believer noted that they valued prayers positively because "someone is acknowledging the hardships I am going through and wishes for me to get through them successfully" (R_161) while a Christian participant explained: as "a Christian, prayer is invaluable and a source of personal comfort through faith" (R_282).

Further, a large majority of Christians (82 percent) believe that the prayer will result in God intervening to ease their emotional pain. While shares are smaller, many Christians also value the prayer because they believe God will help materially (36 percent) or improve their health (55 percent). Such expectations appear to be misplaced, given previous research shows that prayers for others have no effect on the recipient's health [23], and therefore might bias the value of prayers upwards. They might also explain why prayers may reduce material aid [8]–if God is expected to intervene materially in response to prayers, the perceived need for material aid may be lower.

We also asked Christians who positively value prayers (N = 375/451) about the probability that the prayer from the stranger would be answered by God. Their average response was 78 percent. Amongst these participants, those who were more religious (as measured by

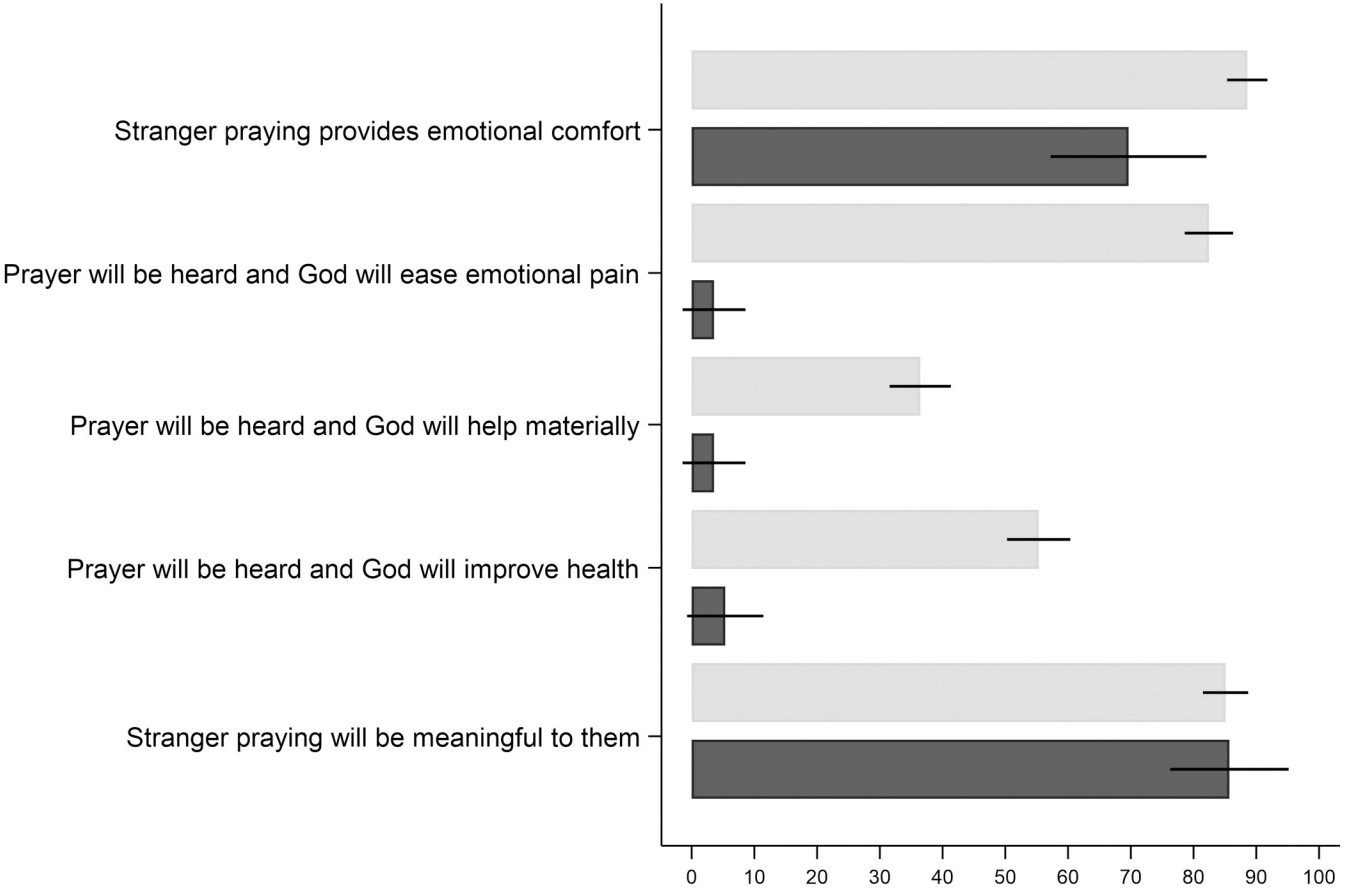

**Fig 2. Share that agrees with factors that contribute to the positive value of the prayer; Christians (light grey) and non-believers (dark grey).** Note: Error bars represent 95% confidence intervals.

frequency of church attendance), Republicans and those with low income (compared to high income) stated a higher probability that God would answer the prayer. For details, see Supplemental Online Material.

The share of non-believers who value the prayer and believe the prayer will result in help from God (whether emotional, material, or health) is not statistically significantly different from zero, i.e., even though some non-believers positively value receiving a prayer, they do not expect the prayer to generate benefits due to divine intervention. Finally, a large majority of both Christians and non-believers positively value the prayer because they think sending the prayer is a meaningful activity for the stranger. Hence, altruism could be an important part of the prayer's value, to both Christians and non-believers–the recipient believes the sender of the prayer will benefit from undertaking the prayer.

While the results shown in Fig 2 indicate why people positively value prayers, it does not show how intensely each factor affects the positive value. Next, we examined the extent to which these factors, and covariates, affect the positive WTP. To do so, we regressed WTP for the prayer from the Christian stranger on agreement with each statement in Fig 2, a set of common demographics—gender, age, conservatism, religious belonging, religiosity (measured as frequency of church attendance) income and college attendance–as well as the type of hardship (issue) described in the experimental survey.

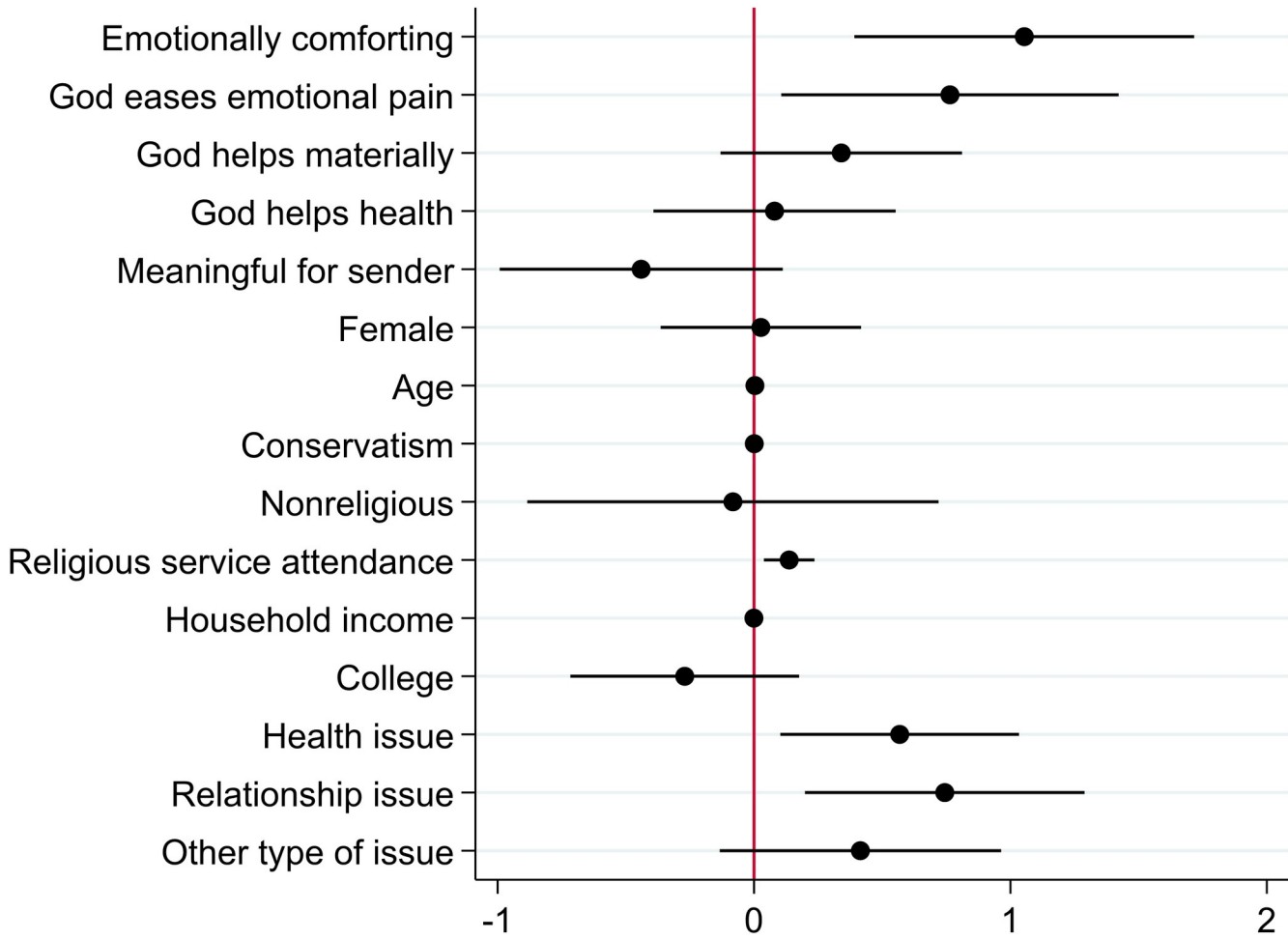

**Fig 3. Determinants of the positive value of prayers.** Note: Coefficients generated by Ordinary Least Squares regression, N = 406, $R^2$ = 0.241. Error bars represent 95% confidence intervals. A value of zero implies that the variable does not affect the average value of the positive WTP for a prayer.

Fig 3 shows that the highest positive value for a prayer is generated if the recipient expects emotional comfort from the prayer. Although Fig 2 shows that many participants value the prayer because it benefits the sender to pray (altruism), the benefit to the sender does not contribute to the average positive value of a prayer (if anything, it brings down the mean positive value of the prayer). Further, beliefs that the prayer generates material help or improved health do not affect the mean positive value of the prayer.

The type of hardship addressed by the prayer also matters to the intensity by which a person values receiving a prayer. Around 30 percent of participants reported a health issue (for self or a loved one) as the hardship, around 30 percent reported a financial issue, between 15 and 20 percent reported a relationship issue, and around 20 percent an issue that does not fall into any of those categories. Recipients value the prayer more if the hardship they experience consists of a health or relationship issue (for themselves or a loved one), compared to if they or a loved one experience a financial issues (the benchmark in the model underlying Fig 3). These results are robust to the inclusion of covariates. Note that while being conservative significantly affects whether a prayer is positively valued (see above), more conservative people who value prayers do not assign a particularly high positive value to the prayer. This result is stable across our measurements of conservatism—it does not matter whether we use the SEC scale (the

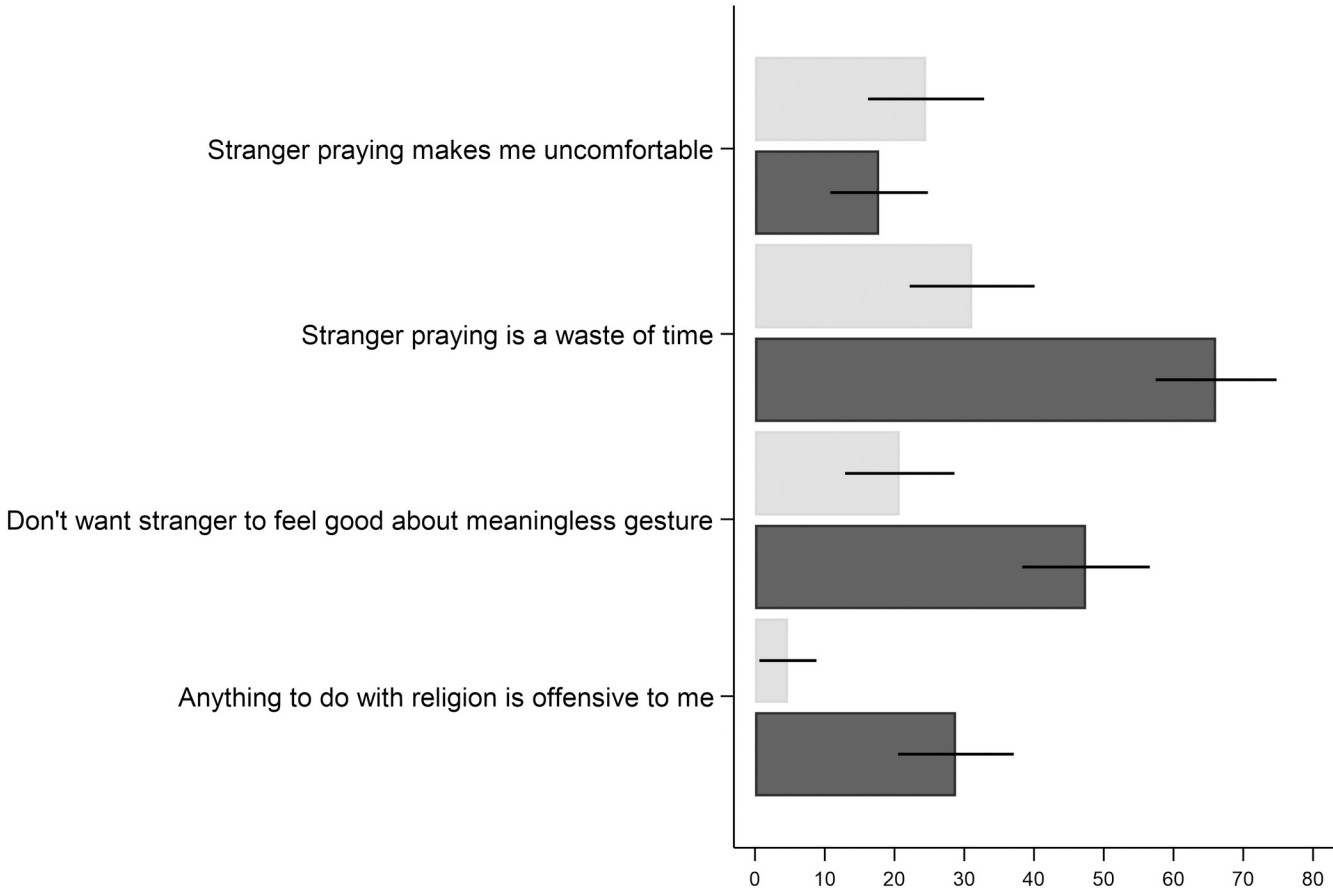

**Fig 4. Share that agrees with factors that contribute to the negative value of the prayer; Christians (light grey) and non-believers (dark grey).** Note: Error bars represent 95% confidence intervals.

conservatism measure in Fig 3), the liberal-conservatism scale or political party belonging as a measure of conservatism.

**Reasons people underline{negatively} value prayers.** Participants who stated a negative WTP for the prayer from the stranger, and were internally consistent in their WTP answers (Christian: N = 106/451; non-believers: N = 118/166), were asked whether the statements in Fig 4 contributed to the negative value of the prayer from the stranger.

The most widespread reason for non-believers to negatively value the prayer is that they do not want the stranger to waste his/her time performing the prayer: 66 percent of non-believers who negatively value prayers state this as a reason. This could potentially be interpreted as an altruistic reason not to value prayers. Further, many non-believers do not want the stranger to feel good about something that is meaningless. For religious Christians who negatively value the prayer, none of the reasons in Fig 4 seems to be a particularly widespread determinant of their negative value. Around 20–30 percent agree that the prayer was negatively valued because it causes them emotional discomfort knowing that a stranger thinks of them, because they do not want the stranger to waste his/her time praying, or because they do not want the stranger to feel good about something that is meaningless.

The open-ended comments point to additional reasons why Christians may negatively value the prayer from a stranger: several participants reported they did not want the prayer because it comes from a stranger, for instance: "do not want anything from someone I do not

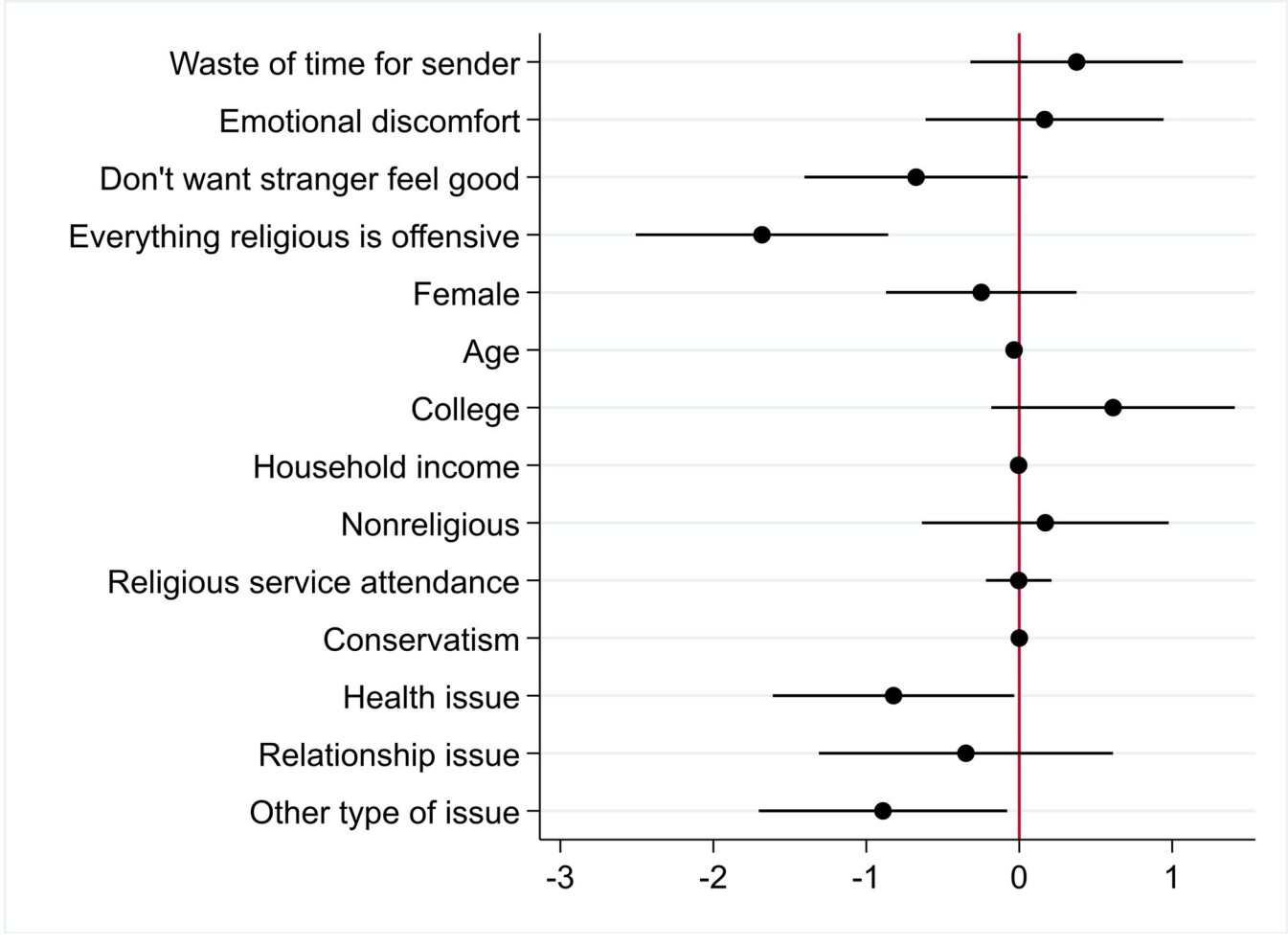

**Fig 5. Determinants of the negative value of prayers.** Note: Coefficients generated by Ordinary Least Squares regression, N = 211, $R^2$ = 0.205. Error bars represent 95% confidence intervals. A value of zero implies that the variable does not affect the average value of the negative WTP for a prayer.

know and don't trust" (R_183) and "I want to be sure he or she is a born again Christian" (R_237). Hence, while many Christians value prayers, others are concerned with who is sending the prayer. Many of those who assigned the prayer a negative value noted they valued prayers but not from strangers. Another self-identified Christian channeled some of the more popular criticisms: "I am weary of hearing about "thoughts and prayers" in response to a crisis, particularly in the case of gun violence" (R_201). Fig 5 explores the intensity by which each factor explains the negative value of prayers. The negative value assigned to receiving a prayer is particularly large if a person is offended by religion. Further, the hardship that the prayer addresses matters. A person who is prayer averse is particularly harmed by receiving the prayer if experiencing a health issue (for self or a loved one), or an issue other than health, relationship or financial, compared to if they or a loved one are experiencing a financial issues (the benchmark in the model underlying Fig 5).

## Conclusions

Beliefs that cause people to positively value prayers are likely to have behavioral implications. Coping and other behaviors are affected by religious beliefs and rituals [24]. We encourage

future research to explore the behavioral impact from experienced and perceived benefits generated by prayers from others. For instance, risk perceptions affect consumption [25], and threats of major hardships affect coping mechanisms [26]. Demand for services that prevent, protect, or insure against risk might therefore differ depending on beliefs about the power of prayers to mitigate risks. Further, it might matter to behavior whether prayers help emotionally, or are expected to benefit health and wealth. When people believe they have God's support they react less to information about potential catastrophes [6]. Future studies may also examine how the values and expectations of prayers are affected by characteristics of both the sender and the prayer, such as social distance between the sender and receiver, perceived closeness to God (e.g., a religious authority, versus a stranger or a friend), and whether the prayer is conducted in private or in public. We also encourage future research to elicit senders' beliefs about the effects of prayers, and examine how such beliefs might guide the choice of activity to aid people in hardships. For instance, prayers may crowd out charity donations [8, 27]. Finally, future studies may add nuance to the findings presented in this study. For instance, the broader beliefs that prayers help with health or materially may mask interesting heterogeneity in the precise meaning of those beliefs across different people. Qualitative research could be particularly helpful in shedding light on such more fine grained beliefs. Further, our analysis relied primarily on participants' agreement to a set of pre-specified statements about what might give prayers value (or not). We complemented these statements with open ended questions about what makes people value (or not) receiving prayers. While the responses to the open ended questions did not indicate that our analysis excluded other relatively important determinants of prayers' value, and were broadly consistent with our pre-specified statements, future work can seek to build on this research and further explore such determinants as well as their relative importance both to senders and receivers of prayers.

Our research suggests avenues for public dialogue that may be helpful in bridging the divisiveness of intercessory prayers as a response to crises in the U.S. In particular, people who send prayers may emphasize both that the gestures are offered as emotional support, and aimed to complement, rather than be in lieu of, material support. Further, signaling that these gestures are intended towards fellow believers may reduce animosity. Senders may further emphasize their devotion when offering prayers to fellow Christians. Also, to maximize the benefits to those in hardship, people who consider showing their support by sending prayers may consider both whom to target with prayers and the type of hardship experienced by the recipient of the prayer. We find that people have particularly strong preferences for or against prayers when these gestures are sent in support of a health or relationship hardship. For instance, while our results showed that those who value prayers assign a particularly high value to receiving a prayer in support of a health hardship, compared to a financial hardship, we also find that those who are prayer averse assign a particularly low (i.e., particularly negative) value to receiving a prayer in support of the same type of hardship.

## Author Contributions

**Conceptualization:** Linda Thunström, Shiri Noy.

**Data curation:** Linda Thunström, Shiri Noy.

**Formal analysis:** Linda Thunström, Shiri Noy.

**Investigation:** Linda Thunström, Shiri Noy.

**Methodology:** Linda Thunström, Shiri Noy.

**Project administration:** Linda Thunström.

**Visualization:** Linda Thunström.

**Writing – original draft:** Linda Thunström.

**Writing – review & editing:** Linda Thunström, Shiri Noy.

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
