## [Decision Letter · Decision Letter 0]

10 Feb 2022

PONE-D-21-23989

What We Think Prayers Do: Americans’ Expectations and Valuation of Intercessory Prayer

PLOS ONE

Dear Dr. Thunstrom,

Thank you for submitting your manuscript to PLOS ONE. After careful consideration, we feel that it has merit but does not fully meet PLOS ONE’s publication criteria as it currently stands. Therefore, we invite you to submit a revised version of the manuscript that addresses the points raised during the review process.

We look forward to receiving your revised manuscript.

Kind regards,

Sonia Brito-Costa, Ph.D.

Academic Editor

PLOS ONE

https://journals.plos.org/plosone/s/fileid=ba62/PLOSOne_formatting_sample_title_authors_affiliations.pdf".

Reviewers' comments:

**Comments to the Author**

5. Review Comments to the Author

Reviewer #1: This paper uses an experimental survey design to demonstrate 1) the differing valuation of receiving prayers and 2) the reasons contributing to this differential valuation. The paper was written in clear and crisp prose and organized in a logical manner. The authors are clearly well-versed in the social scientific literature and experimental / quantitative methods. My recommendation is to accept this paper. I believe the recommendations below would improve the quality of the paper, but their necessity for publication is left to the editors discretion.

p4 - Please elaborate on the "data quality checks" that Qualtrics performs to avoid "data quality issues."

p6-7 - Please elaborate on the origin of "the set of statements about factors that might have determined their value of the prayer." These play an important role in subsequent regression analysis / conclusions (i.e. Fig 2 and 4), but because the study was designed to explore why participants valued prayer, I figured these factors would be identified by the participants not the researchers. Was there a relationship between the open-ended responses and these predetermined factors? In an ideal world, one would construct the set of statements after first analyzing the open-ended responses, etc.

p9 - Using simple survey methods for something as complicated as religious belief / theology has always been difficult, and I couldn't help but think about how liberal theologians or sociologists (or survey respondents) might "translate" or interpret different claims. Paul Tillich thinks of God as being-itself, so the phrase "God will improve health" would have a much different meaning. Durkheim's idea that God is a symbol of society would mean that "God will ease emotional pain" and "stranger praying provides emotional comfort" would almost be synonymous. I don't have a solution to this issue.

Fig 3 and 5 are both great examples of presenting regressions, but they of course lose some of the information given by traditional tables. What kind of regression was run? What type of model diagnostics should we be aware of? Something like an R-squared or measure of overall fit should be included.

Congratulations on an insightful paper and best of luck to the authors.

---

## [Author Response · Author response to Decision Letter 0]

4 Mar 2022

Dear Dr Brito-Costa,

Thank you for the opportunity to revise our manuscript. We are excited about the prospect of our study being published in PLOS ONE. We are also grateful to both you and the Reviewer for taking the time and effort to provide insightful comments on our manuscript. Please find below our point-by-point response to the Reviewer.

In addition to editing the manuscript to accommodate the helpful comments by the Reviewer, we have:

1. Deposited the data, code, and survey instrument in the open repository ICPSR. We have added the following information to the manuscript (p.7):

The survey instrument is deposited, as part of the Supplemental online material, in the open repository ICSPR at https://www.openicpsr.org/openicpsr/project/163981/version/V1/view. There, we also post the data and code used for all analysis in the manuscript. 

2. Reviewed the reference list so ensure that it is complete and correct, as well as edited to PLOS ONE style.

3. Ensured the manuscript meets PLOS ONE’s style requirements.

4. Corrected a typo in the note under Figure 5. It previously read “Note: Error bars represent 95% confidence intervals. A value of zero implies that the variable does not affect the average value of the positive WTP for a prayer.” This has been edited to: “…negative WTP…”

5. Deleted a repetition of a word to increase the flow of the sentence in the second paragraph of the Introduction. Two consecutive sentences started with “In particular,…” We therefore deleted those two words in the beginning of the second sentence.

6. Noted that we were inconsistent in using “n” or “N” to denote sample sizes, so we edited to “N” throughout the manuscript. 

7. We have added the following reference – a reference that we were not aware of when writing the previous version of the manuscript, but that is appropriate for us to cite (note that the below study does a very different analysis than the one presented in our paper, it just adds a reference on a theme that is already mentioned in our concluding discussion):

[27] Greenway TS, Schnitker, SA, Shepherd, AM. Can prayer increase charitable giving? Examining the effects of intercessory prayer, moral intuitions, and theological orientation on generous behavior. The International Journal for the Psychology of Religion. 2018; 28(1): 3-18.

Reviewer comments:

We much appreciate the thoughtful comments from the Reviewer and the helpful suggestions to add important details to the paper. Below, we respond to each comment in turn.

This paper uses an experimental survey design to demonstrate 1) the differing valuation of receiving prayers and 2) the reasons contributing to this differential valuation. The paper was written in clear and crisp prose and organized in a logical manner. The authors are clearly well-versed in the social scientific literature and experimental / quantitative methods. My recommendation is to accept this paper. I believe the recommendations below would improve the quality of the paper, but their necessity for publication is left to the editors discretion.

p4 - Please elaborate on the "data quality checks" that Qualtrics performs to avoid "data quality issues."

We thank the Reviewer for pointing out that more elaboration on the data quality checks by Qualtrics would be helpful to the reader, and have added the following (p.4-5):

For instance, to ensure validity of the responses and avoid duplication, Qualtrics checks IP address of all responses and uses digital fingerprinting technology. Qualtrics replaces respondents that fail any attention checks, as defined by the researcher, or in other ways appear fraudulent (this is evaluated in collaboration with the researcher), as well as respondents who seem to rush through the survey, i.e., completes the survey in less than half the median survey completion length. Verification of responder identities is generally done by Qualtrics’ sample partners and include TrueSample, Verity, SmartSample, panelist ID number, cookies, Geo-IP address, LinkedIn information comparison, and digital fingerprinting. 

p6-7 - Please elaborate on the origin of "the set of statements about factors that might have determined their value of the prayer." These play an important role in subsequent regression analysis / conclusions (i.e. Fig 2 and 4), but because the study was designed to explore why participants valued prayer, I figured these factors would be identified by the participants not the researchers. Was there a relationship between the open-ended responses and these predetermined factors? In an ideal world, one would construct the set of statements after first analyzing the open-ended responses, etc.

We thank the Reviewer for this important comment. As noted by the Reviewer, we elicited participants’ open-ended responses and their responses to our set of constructed statements in the same survey, i.e., we did not base our statements on the open-ended responses. Rather, our statements were based on findings from previous studies discussed in the Introduction. Our aim was for our analysis to rely primarily on the set of statements we provided, while gaining potentially important additional information from the open-ended questions. 

The responses to our statements are, however, quite consistent with the responses to the open-ended questions. Our preliminary analysis of the open-ended questions suggests that the majority (about 60%) of open ended responses fall, in a general way, within the categories we pre-specified in the statements. By ‘general,’ we mean that there is some subjectivity in how to categorize the open ended responses. For instance, we classified an open ended response of “The value of any kind of support, be it prayer or good thoughts or good wishes, are more valuable than $5 or less” as being consistent with the prayer offering emotional support, while we classified the response “Because I believe it will cause a positive outcome” as consistent with our pre-specified category of God can help materially (that is, change circumstances). However, these classifications undoubtedly can be questioned – our attempt to explore the dimensions of the open-ended questions is rough and the qualitative data is more “open” than our quantitative data, making it harder to analyze in a stringent manner. 

There were other open-ended responses that did not answer the question, i.e., did not provide information on why prayers were valued (about 15%), but rather re-iterated that prayers were valued, such as “Everything helps so I would welcome the prayer” or “I think we should take all the prayers we can get. They never hurt!” Yet others (about 15%) were inconsistent with their elicited values of prayers. For instance, some people who valued prayers answered the open-ended question as “I do not believe in a god nor prayers and I won't compromise my lack of belief in a supreme power” or “I do not believe that prayer will supply a resolution.” or “I am an agnostic & have been many years.” 

In sum, given our analytical approach, we believe pre-specifying the reasons for valuing receiving prayers (or not), as we did in the set of statements faced by responders (which in turn were based broadly on previous literature that indicates reasons for valuing prayers for self), where respondents can agree or disagree or remain neutral, generated an informative overview of the kinds of reasons for people to value receiving prayers (or not). Importantly, the open ended responses also did not indicate that our statements missed some important reason for respondents’ values. However, we fully agree with the Reviewer that more qualitative methods could provide important further insights. Further, we hope future research can further explore this topic.

We have added the following information to the Conclusions of the manuscript about the responses to our open-ended questions (p. 16-17):

Further, our analysis relied primarily on participants’ agreement to a set of pre-specified statements about what might give prayers value (or not). We complemented these statements with open ended questions about what makes people value (or not) receiving prayers. While the responses to the open ended questions did not indicate that our analysis excluded other relatively important determinants of prayers’ value, and were broadly consistent with our pre-specified statements, future work can seek to build on this research and further explore such determinants as well as their relative importance both to senders and receivers of prayers.

p9 - Using simple survey methods for something as complicated as religious belief / theology has always been difficult, and I couldn't help but think about how liberal theologians or sociologists (or survey respondents) might "translate" or interpret different claims. Paul Tillich thinks of God as being-itself, so the phrase "God will improve health" would have a much different meaning. Durkheim's idea that God is a symbol of society would mean that "God will ease emotional pain" and "stranger praying provides emotional comfort" would almost be synonymous. I don't have a solution to this issue.

We thank the Reviewer for this thoughtful comment. We agree that measuring religion and theology is indeed a very difficult undertaking. We also agree that respondents may differently interpret the statements in our survey. Qualitative methods may provide further nuance to our results. To address this issue, we have added the following to the Conclusions (p.16):

Finally, future studies may help add nuance to the findings presented in this study. For instance, the stated beliefs in our study that prayers help materially, or with health, may mask interesting heterogeneity in the more precise meaning of those beliefs across different people. Qualitative research could be particularly helpful in shedding light on such more fine grained beliefs.

Fig 3 and 5 are both great examples of presenting regressions, but they of course lose some of the information given by traditional tables. What kind of regression was run? What type of model diagnostics should we be aware of? Something like an R-squared or measure of overall fit should be included.

We thank the Reviewer for noting that important information about the estimated models underlying Figures 3 and 5 was missing. We have added information on the type of regression model used (Ordinary Least Squares Regression; OLS), the number of observations in each regression, and the R-squared measures. 

Specifically, we have added the following information to the note directly under Figure 3:

Coefficients generated by Ordinary Least Squares regression, N=406, R2= 0.241. 

And the following information to the note directly under Figure 5:

Coefficients generated by Ordinary Least Squares regression, N=211, R2= 0.205.

---

## [Editor Report · Decision Letter 1]

9 Mar 2022

What We Think Prayers Do: Americans’ Expectations and Valuation of Intercessory Prayer

PONE-D-21-23989R1

Dear Dr. Thunstrom,

We’re pleased to inform you that your manuscript has been judged scientifically suitable for publication and will be formally accepted for publication once it meets all outstanding technical requirements.

Kind regards,

Sonia Brito-Costa, Ph.D.

Academic Editor

PLOS ONE
---

## [Editor Report · Acceptance letter]

11 Mar 2022

PONE-D-21-23989R1 

What We Think Prayers Do: Americans’ Expectations and Valuation of Intercessory Prayer 

Dear Dr. Thunström:

I'm pleased to inform you that your manuscript has been deemed suitable for publication in PLOS ONE. Congratulations! Your manuscript is now with our production department. 

Kind regards, 

on behalf of

Dr. Sonia Brito-Costa 

Academic Editor

PLOS ONE